# Association of Mutation Profiles with Postoperative Survival in Patients with Non–Small Cell Lung Cancer

**DOI:** 10.3390/cancers12113472

**Published:** 2020-11-21

**Authors:** Taichiro Goto, Kei Kunimasa, Yosuke Hirotsu, Takahiro Nakagomi, Yujiro Yokoyama, Rumi Higuchi, Sotaro Otake, Toshio Oyama, Kenji Amemiya, Hitoshi Mochizuki, Masao Omata

**Affiliations:** 1Lung Cancer and Respiratory Disease Center, Yamanashi Central Hospital, Yamanashi 400-8506, Japan; nakagomi.takahiro@keio.jp (T.N.); Yujiro.yokoyama@sluhn.org (Y.Y.); r-iguchi1504@ych.pref.yamanashi.jp (R.H.); ootake-bdcg@ych.pref.yamanashi.jp (S.O.); 2Genome Analysis Center, Yamanashi Central Hospital, Yamanashi 400-8506, Japan; kei.kunimasa@oici.jp (K.K.); hirotsu-bdyu@ych.pref.yamanashi.jp (Y.H.); amemiya-bdcd@ych.pref.yamanashi.jp (K.A.); h-mochiduki2a@ych.pref.yamanashi.jp (H.M.); m-omata0901@ych.pref.yamanashi.jp (M.O.); 3Department of Thoracic Oncology, Osaka International Cancer Institute, Osaka 541-8567, Japan; 4Department of Pathology, Yamanashi Central Hospital, Yamanashi 400-8506, Japan; t-oyama@ych.pref.yamanashi.jp; 5Department of Gastroenterology, The University of Tokyo Hospital, Tokyo 113-8655, Japan

**Keywords:** lung cancer, next generation sequencing, mutation, *TP53*, survival

## Abstract

**Simple Summary:**

In this study, we comprehensively and synthetically analyzed mutations in lung cancer based on the next generation sequencing data of lung tumors surgically removed from the patients, and identified the mutation-related factors that can affect clinical outcomes. Detailed understanding of the genomic landscape of lung cancers will establish the ideal model for best surgical outcomes in the era of “precision medicine”.

**Abstract:**

Findings on mutations, associated with lung cancer, have led to advancements in mutation-based precision medicine. This study aimed to comprehensively and synthetically analyze mutations in lung cancer, based on the next generation sequencing data of surgically removed lung tumors, and identify the mutation-related factors that can affect clinical outcomes. Targeted sequencing was performed on formalin-fixed paraffin-embedded surgical specimens obtained from 172 patients with lung cancer who underwent surgery in our hospital. The clinical and genomic databases of the hospital were combined to determine correlations between clinical factors and mutation profiles in lung cancer. Multivariate analyses of mutation-related factors that may affect the prognosis were also performed. Based on histology, *TP53* was the driver gene in 70.0% of the cases of squamous cell carcinoma. In adenocarcinoma cases, driver mutations were detected in *TP53* (26.0%), *KRAS* (25.0%), and epidermal growth factor receptor (*EGFR*) (23.1%). According to multivariate analysis, the number of pathogenic mutations (≥3), presence of a *TP53* mutation, and *TP53* allele fraction >60 were poor prognostic mutational factors. The *TP53* allele fraction tended to be high in caudally and dorsally located tumors. Moreover, *TP53*-mutated lung cancers located in segments 9 and 10 were associated with significantly poorer prognosis than those located in segments 1–8. This study has identified mutation-related factors that affect the postoperative prognosis of lung cancer. To our knowledge, this is the first study to demonstrate that the *TP53* mutation profile varies with the site of lung tumor, and that postoperative prognosis varies accordingly.

## 1. Introduction

Along with the technological advancement in next generation sequencing (NGS), accumulated findings on mutations, associated with lung cancer, have led to the development of mutation-based precision medicine [1]. In fact, novel therapies, based on information regarding cancer antigens and cancer mutations, such as immune checkpoint blockade and molecular-targeted therapy, have recently been developed, and treatment outcomes for lung cancer, have improved dramatically [2,3]. Therefore, patient-based clinicogenomic datasets may significantly accelerate the advancement of clinical practice and the development of novel therapeutics.

Although, the postoperative prognosis of lung cancer has been conventionally and stochastically predicted, based on the histological classification and the tumor-node-metastasis (TNM) stage [4,5], a prognostic model specifically applicable for each case has not been established. The criteria for adjuvant chemotherapy are also not clear. Hence, accurate criteria for adjuvant chemotherapy based on appropriate prognostic models in the future should be urgently established [6,7].

Our study group has continuously analyzed the NGS data of patients with lung cancer since January 2014. Hence, this study aimed to synthetically analyze the correlation between mutation profiles of patients with lung cancer and clinical factors, using the integrated results of this NGS analysis, and identify mutational factors affecting clinical outcomes.

## 2. Results

### 2.1. Patient Characteristics

We studied surgical samples from 172 patients with lung cancer who underwent surgery at our hospital between June 2014 and June 2019. The characteristics of the enrolled patients are summarized in Table 1. Among the 172 patients, 116 were men and 56 were women, and 137 were smokers and 35 were non-smokers. The age of patients ranged between 44 and 90 (mean ± SD, 71.1 ± 10.8) years. Histologically, there were 103 cases of adenocarcinoma, 40 cases of squamous cell carcinoma, 4 cases of adenosquamous carcinoma, 10 cases of small cell carcinoma, and 15 cases with other histology. In terms of pathological stage, there were 79 stage I, 21 stage II, 33 stage III, and 39 stage IV cases. One patient received four cycles of preoperative cisplatin + vinorelbine chemotherapy. The median (range) follow-up period for all censored cases was 973 (21–2056) days.

### 2.2. Clinicogenomic Features and Associations

In total, 2372 mutations were detected at an allele fraction (AF) of >1%, and the mean number of mutations per cancer lesion was 13.8 ± 2.9. Of these, 447 mutations were annotated as pathogenic (or oncogenic) mutation (Appendix A), and the mean number of pathogenic mutations per cancer lesion was 2.6 ± 0.4. The frequency of short variant mutations is shown in Figure 1. The most common mutation was in the gene encoding tumor protein p53 (*TP53*), which was detected in approximately 60% of patients with lung cancer. When only pathological mutations at an AF of >20% were examined, common mutations were detected in genes encoding *TP53*, *KRAS*, and *EGFR* in adenocarcinoma and in the *TP53* gene in squamous cell carcinoma (Figure 2A,B). Pathogenic mutations with AF of >20 were regarded as driver mutations, and 89.5% of patients with lung cancer were found to harbor driver mutations in *EGFR, KRAS*, or *TP53*, which were found to be the three major mutations in lung cancer. Fusion gene screening through transcriptome sequencing revealed that *ROS1*, *ALK*, and *RET* fusion genes were not detected in any of the 172 patients who enrolled in this study.

### 2.3. Pathogenic vs. Nonpathogenic Mutations in EGFR, KRAS, and TP53

For *EGFR* and *TP53*, the number of pathogenic mutations was 1.24 and 1.38 times higher than the number of nonpathogenic mutations, respectively, whereas, for *KRAS*, the former was 7.40 times higher and significantly higher than the latter (Appendix A). In other words, most mutations detected in *KRAS* were pathogenic mutations.

### 2.4. Multivariate Analysis of Progression-Free Survival (PFS) after Surgery

When multivariate analysis was performed with age, gender, histology, stage, number of pathogenic mutations, presence of *EGFR*, *KRAS*, and *TP53* mutations, and AF of *EGFR*, *KRAS*, and *TP53* mutations to identify factors affecting PFS, the stage, number of pathogenic mutations, presence of *TP53* mutations, and AF of *TP53* mutations were found to be significant prognostic factors (*p* < 0.05).

### 2.5. Number of Oncogenic Mutations in Lung Cancer and Patient Prognosis

The largest proportion of patients (67/172 patients) harbored one oncogenic mutation. The mean number of oncogenic mutations per tumor was 2.6 ± 0.4 (Figure 3A). The prognosis was significantly poorer in tumors with three or more oncogenic mutations than in tumors with two or less mutations (Figure 3B).

### 2.6. Presence of TP 53 Mutation and Patient Prognosis

Irrespective of oncogenic and non-oncogenic mutations, patients with *TP53* mutations had significantly poorer prognosis than those without *TP53* mutations (Figure 4A). Next, we compared the prognosis among patients with different numbers of *TP53* mutations, including oncogenic and non-oncogenic mutations. Prognosis was significantly poorer in patients with tumors harboring 0, 1, and 2 or more mutations (in this order) (Figure 4B).

### 2.7. Allele Fraction of TP 53 Mutation and Patient Prognosis

According to AF, *TP53* mutations, including oncogenic and non-oncogenic mutations, were classified into three groups: AF = 0, 0 < AF < 60, and AF > 60. The *TP53* mutation with AF = 0 represented cancer without any *TP53* mutations. In tumors with multiple *TP53* mutations, the highest AF was used to classify each tumor. PFS was significantly shorter in the group of *TP53* mutations with AF > 60 than in the two other groups (hazard ratio (HR) 2.84, 95% confidence interval (CI) 1.48–5.48, compared to AF = 0; HR 2.19, 95% CI 1.05–4.78, compared to 0 < AF < 60) (Figure 4C).

### 2.8. Mutation Profiles According to Cancer Location

We investigated whether the mutation profiles differed with tumor sites. Between tumors located at the cranial side (segments 1, 2, 3, and 1 + 2, *n* = 84) and the caudal side (segments 7, 8, 9, and 10, *n* = 56), no correlation was observed in the presence or absence of *EGFR*, *KRAS*, or *TP53* mutations (Appendix A). No correlation was also observed in the presence or absence of *EGFR*, *KRAS,* or *TP53* mutations between tumors located at the ventral side (segments 3, 4, and 5, *n* = 29) and the dorsal side (segments 2, 6, 9, and 10, *n* = 59) (Appendix A). These findings revealed that tumor sites were not associated with the presence or absence of specific gene mutations.

No significant differences were observed in the AFs of *EGFR* or *KRAS* mutations when the AFs of *TP53*, *EGFR*, and *KRAS* mutations were compared between tumor sites. However, the AFs of *TP53* mutations were significantly higher in tumors located at the caudal and dorsal sides (Figure 5A,B).

### 2.9. Characteristics of TP53-Mutated Lung Cancer in Segments 9 and 10

Among *TP53*-mutated lung cancers, AFs of *TP53* mutations, including oncogenic and non-oncogenic mutations, were significantly higher in lung cancers located in segments 9 and 10, the dorsal inferior lobes, than in those located in the other segments (1 to 8) (Figure 6A, Appendix A). In addition, among *TP53*-mutated lung cancers, PFS was significantly shorter in patients with lung cancers located in segments 9 and 10 than in those where tumors were located in the other segments (1 to 8) (Figure 6B). In contrast, among lung cancers without *TP53* mutations, no significant difference was observed in postoperative prognosis between lung cancers located in segments 9 and 10 and those located in segments 1 to 8 (Appendix A). Among patients with *TP53*-mutated lung cancer, no significant differences in either the proportion of smokers (segments 1–8: 86.8%, segments 9–10: 82.1%, *p* = 0.55) or the smoking index, defined as number of cigarettes smoked per day multiplied by years smoked (segments 1–8: 280 ± 54, segments 9–10: 234 ± 62, *p* = 0.83), were observed between those with lung cancers located in segments 9 and 10 and those with lung cancers located in segments 1 to 8.

### 2.10. Correlation between Smoking and TP53 Mutations

To further investigate the correlation between smoking and *TP53* mutations, we compared the prevalence of *TP53* mutations between smokers (*n* = 137) and non-smokers (*n* = 35) with lung cancer. The results showed a significantly higher prevalence of *TP53* mutations in smokers with lung cancer (smokers: 65.0% versus non-smokers: 42.9%, *p* = 0.018). Comparison of the AFs of *TP53* mutations between smokers and non-smokers with lung cancer revealed that the AFs were significantly higher in smokers with lung cancer (smokers: 32.1 ± 3.1 versus non-smokers: 12.5 ± 5.6, *p* = 0.003).

## 3. Discussion

In lung cancer treatment, gene sequencing has been performed mainly as a companion diagnostic approach for identifying mutations in genes such as *EGFR* and anaplastic lymphoma kinase (*ALK*) and subsequent selection of molecular target drugs [8,9,10]. Currently, targeted sequencing using NGS is being increasingly performed at clinical facilities worldwide [1,11]. The advancement in NGS technology has realized comprehensive detection of many gene mutations in lung cancer. However, the correlation between the detected mutation profiles and the pathology of lung cancer has not been sufficiently analyzed. The availability of NGS is still limited, due to its cost, cumbersome device operations and its supposedly limited clinical benefits. Furthermore, even in clinical facilities performing NGS, the amount of data generated at each facility is not sufficient for scientific analysis. In this study, we aimed to analyze the effect of gene mutation profiles on the prognosis of lung cancer. Toward this objective, examination of many surgical samples collected from a large patient cohort with similar clinical characteristics was required for effective analysis of prognosis. As gene sequencing on surgical samples is being aggressively performed at our hospital since July 2014, we had access to sufficient number of sequenced samples [12,13,14,15,16,17,18,19]. In this context, association between clinical and genomic characteristics was analyzed in this study. Among the mutational factors, the number of pathogenic mutations, presence of *TP53* mutation, and AF of *TP53* mutation were found to be the prognostic factors.

Our study revealed that, on average, one cancer lesion harbors 2.6 pathogenic mutations, which can be explained by branched evolution model [20]. Furthermore, several studies have suggested that even if one mutation pathway is inhibited, the presence of other functional pathways of driver mutations can drive proliferation of tumor cells and render the cancer aggressive [9,21,22,23]. Therefore, when multiple driver mutations are incorporated into cancer during the course of tumor evolution, the proliferation drive is activated at multiple stages, which enhances the aggressiveness of cancer. Our study also showed that PFS was shorter in tumors with a large number of pathogenic mutations, validating the above-mentioned hypothesis from another perspective. For lung cancer harboring multiple pathogenic mutations, radical treatment may be recommended considering tumor aggressiveness. Meanwhile, cases of lung cancer harboring no oncogenic mutation include, (i) those harboring mutations that cannot be sequenced via NGS for reasons, such as presence of fusion genes, DNA amplification, DNA methylation, and histone modification; (ii) those harboring mutations located at gene sites other than the targeted regions, which can be sequenced on our cancer panel.

The correlation between the presence of *TP53* mutations and poor postoperative prognosis has been suggested by previous studies involving immunostaining [24,25,26]. Our study validated this observation. However, owing to low specificity, immunostaining can only detect the presence or absence of abnormal TP53 proteins, but does not directly detect *TP53* mutations [27]. Furthermore, it does not reflect the details of *TP53* mutations (e.g., mutational pattern and AF) [28]. In relation to the prognostic effects of *TP53* mutations on non-small cell lung cancer (NSCLC) detected via direct sequencing, inconsistent results have been reported previously. Aisner et al. [29] reported that *TP53* mutations are associated with shorter survival. Whereas, Devarakonda et al. [30] and Ma et al. [31] reported that *TP53* mutations exhibited no such prognostic effect. Meanwhile, some other studies have revealed that *TP53* mutations frequently exhibit intra-tumor heterogeneity (ITH) in NSCLC [32,33]. Based on this finding, Lee et al. [34] reported that *TP53* mutations with ITH exhibit worse survival when compared to wild-type *TP53*, whereas *TP53* mutations without ITH did not exhibit such an extreme phenotype. Our study, in which *TP53* mutations were directly detected by deep targeted sequencing, revealed that not only the number, but also the AF of *TP53* mutations correlated with prognosis. Therefore, we clearly demonstrated that *TP53* mutations are additively involved in tumor aggressive behavior. In our study, tumor cells were selected using laser capture microdissection (LCM) before DNA extraction. Although, measurement bias is inevitable for AF measurement because of the effect of contamination with normal cells, a single technician (Kenji Amemiya) has performed LCM as uniformly as possible at our institute. The result of this analysis, showing that AF of *TP53* mutations affects prognosis, is an interesting and novel finding in oncology.

The hypothesis that aggressiveness of lung cancer may vary with tumor sites has not been verified scientifically till date. In our study, AFs of *TP53* mutations were significantly high in tumors located at the dorsal and caudal sides, and the prognosis of *TP53*-mutated lung cancer located in segments 9 and 10 was poor. To date, there has been no reports on the effect of the sites of lung cancer on prognosis. This is possibly the first report providing scientific verification of this effect. Interestingly, the common sites of lung cancer with poor prognosis, the dorsal and caudal regions of the lung, corresponded to the common sites of idiopathic pulmonary fibrosis. The dorsal and caudal regions of the lung may be vulnerable to smoke. Owing to the lack of significant difference in the smoking status and the smoking index between patients with *TP53*-mutated lung cancer located in segments 9 and 10 and that located in segments 1 to 8, we concluded that even a comparable amount of smoking may exert a strong effect on cancer initiation and progression in segments 9 and 10 in the dorsal inferior lobe. Detailed investigation regarding the association between tumor sites and carcinogenesis associated with *TP53* mutations is warranted. In the future, survival after surgery may be improved by changing treatment strategies and the extent of resection based on the mutations harbored and the location of the tumors.

A limitation of this study is that it is a single-institution study with a small sample size. Furthermore, unlike whole exome sequencing, panel sequencing does not provide information regarding all exons, and thus, it is possible that each tumor may actually possess much more pathogenic mutations. However, the lung cancer panel that we developed covers the main hot spots of gene mutations in lung cancer, and we assume that almost all significantly mutated genes associated with lung cancer can be detected. In addition, as the objective was to investigate the postoperative prognosis, we did not investigate the correlation between mutations and responses to medical treatment (drug therapy). Currently, the development of novel therapeutic strategies for lung cancer is rapidly progressing. In the future, various therapeutics, in addition to *EGFR*-tyrosine kinase inhibitors, *ALK*-inhibitors, programmed cell death 1 antibody, and cytotoxic T lymphocyte antigen-4 antibody, will be developed [2,35,36,37]. The correlation between these novel therapies and presence of mutations should be determined to develop precision medicine. Moreover, the treatment for postoperative recurrence has to be substantially changed to improve both PFS and overall survival. The establishment of a large-scale clinicogenome database at the international level, and the promotion of large-scale clinical studies using such a database, are warranted.

## 4. Methods

### 4.1. Patients and Sample Preparation

The study involved 172 patients who underwent surgery in our Department between June 2014 and June 2019. These patients provided written informed consent for the genetic research studies, which were performed in accordance with protocols approved by the Institutional Review Board of Yamanashi Central Hospital (approval date is 21 May 2014, although no number or any ethics code was allocated). Patient data were collected from our institutional cancer registry database and from patient follow-up visits to our outpatient office. The information collected from the records included preoperative patient characteristics, disease status, operative procedure, pathological diagnosis, and follow-up data. Histological typing was performed according to the World Health Organization classification [38], and cancer staging was based on the TNM classification of the International Union Against Cancer, 8th edition [39].

The serial section of formalin-fixed, paraffin-embedded (FFPE) tissue was stained with hematoxylin-eosin and then micro-dissected using an ArcturusXT laser capture microdissection system (Thermo Fisher Scientific, Tokyo, Japan). Tumor DNA was extracted using the QIAamp DNA FFPE tissue kit (Qiagen, Tokyo, Japan). DNA fragmentation in FFPE DNA was assessed using The TaqMan RNase P Detection Reagents Kit and the FFPE DNA QC Assay on ViiA7 Real-Time PCR instrument (Thermo Fisher Scientific). Human control genomic DNA (included with TaqMan RNase P Detection Reagents Kit) was serially diluted 4 times for a 5-point for a standard curve and the absolute DNA concentrations were determined. Assessment of DNA fragmentation was estimated with the ratio of DNA (relative quantification, RQ) obtained for the long amplicon (256 bp) to the short amplicon (87 bp). RQ value was an indicator of the degradation level of genomic DNA. A peripheral blood sample was drawn from each patient immediately prior to surgery and was collected into EDTA-2Na tubes. The buffy coats were isolated following centrifugation of these samples at 820× *g* at 25 °C for 10 min. DNA was extracted from the buffy coats using the QIAamp DNA blood mini kit (Qiagen, Tokyo, Japan).

### 4.2. Targeted Deep Sequencing and Data Analysis

A panel targeting the exons of 53 lung cancer-associated genes (see Appendix A) was selected to perform targeted sequencing. We searched the literature and selected these genes based on the following criteria: (a) Genes often involved in lung cancer reported in The Cancer Genome Atlas [40] and other projects; or (b) genes frequently mutated in lung cancer in the Catalogue of Somatic Mutations in Cancer (COSMIC) database [41]. The primers for targeted sequencing were designed to cover the hot-spot mutations present in the lung cancer-associated genes, using the Ion AmpliSeq designer software (Thermo Fisher Scientific), as reported previously [42,43,44,45,46,47,48]. Sequencing libraries were prepared using Ion AmpliSeq Library Kit Plus (Thermo Fisher Scientific), according to the manufacturer’s instruction. Multiplex PCR was performed with two primer pools. After PCR reaction, primer sequences were partially digested with 2 μL of FuPa reagent (Thermo Fisher Scientific), and then barcoded using Ion Xpress Barcode Adapters (Thermo Fisher Scientific). The library samples were purified using Agencourt AMPure XP reagent (Beckman Coulter, Brea, CA, USA) and subsequently quantified using Ion Library Quantitation Kit (Thermo Fisher Scientific). Each library was diluted to 60 pM, and the same amount of libraries was pooled for one sequence reaction. Emulsion PCR and chip loading were performed on the Ion Chef with the Ion PI Hi-Q Chef kit. Sequencing was performed using the Ion PI Hi-Q Sequencing Kit on the Ion Proton Sequencer (Thermo Fisher Scientific).

The sequence data were processed using standard Ion Torrent suite software running on the Torrent server version 4.4 (Thermo Fisher Scientific). Raw signal data were analyzed using Torrent suite version 4.0. The pipeline included signal processing, base calling, quality score assignment, read alignment to the human genome 19 reference (hg19), and quality control of mapping and coverage analysis. Following data analysis, annotation of single nucleotide variants, insertions, and deletions were performed using the Ion Reporter server system (Thermo Fisher Scientific), and lymphocytes from peripheral blood DNA were used as controls for detecting any variants (tumor-normal pair analysis). We used the following filtering parameters for variant calling on Ion Reporter Local Server: (i) the minimum number of variant allele reads was ≥10, (ii) the coverage depth was ≥50, (iii) variant allele fraction (AF) ≥ 0.01, (iv) UCSC Common SNPs = Not In, (v) Confident Somatic Variants = In. Sequence data were visually confirmed using the Integrative Genomics viewer. The allele fractions represent the number of mutant reads divided by the total number of reads (coverage) at a specific genomic position. The Functional Analysis through Hidden Markov Models prediction in the COSMIC database was used to estimate oncogenic function of single nucleotide variations [49]. In order to estimate oncogenic function of indel mutations, OncoKB, a comprehensive and curated precision oncology knowledge base, was utilized [50]. Thus, all detected mutations were annotated by either COSMIC or OncoKB database [49].

### 4.3. Statistics

Continuous variables were presented as mean ± standard deviation (SD), and compared using unpaired Student’s *t* tests. One-way analysis of variance and the Tukey-Kramer multiple comparison test were used to detect significant differences between groups. Chi-square tests were used to compare the categorical data between groups. Progression-free survival was defined as the period from the day of operation to the day of recurrence or the day of final follow-up. Survival was assessed using the Kaplan-Meier method, and comparisons among the survival curves were made using the log-rank test. To determine the predictors of survival within the cohort, we constructed Cox proportional hazards model including each variable of interest. Multivariate analyses were performed using the JMP function of the SAS software (JMP 15.1.0, SAS Institute, Cary, NC, USA). *p*-values less than 0.05 in the two-tailed analyses were considered to denote statistical significance.

## 5. Conclusions

The number of pathogenic mutations, presence of *TP53* mutations, and AF of *TP53* mutations were identified as mutation factors affecting postoperative prognosis of lung cancer. Importantly, the *TP53* mutation profiles varied with the site of lung cancer, and the postoperative prognosis changed accordingly. Developing a detailed understanding of the genomic landscape of lung cancers will establish the ideal model for best surgical outcomes.

## Figures and Tables

**Figure 1 cancers-12-03472-f001:**
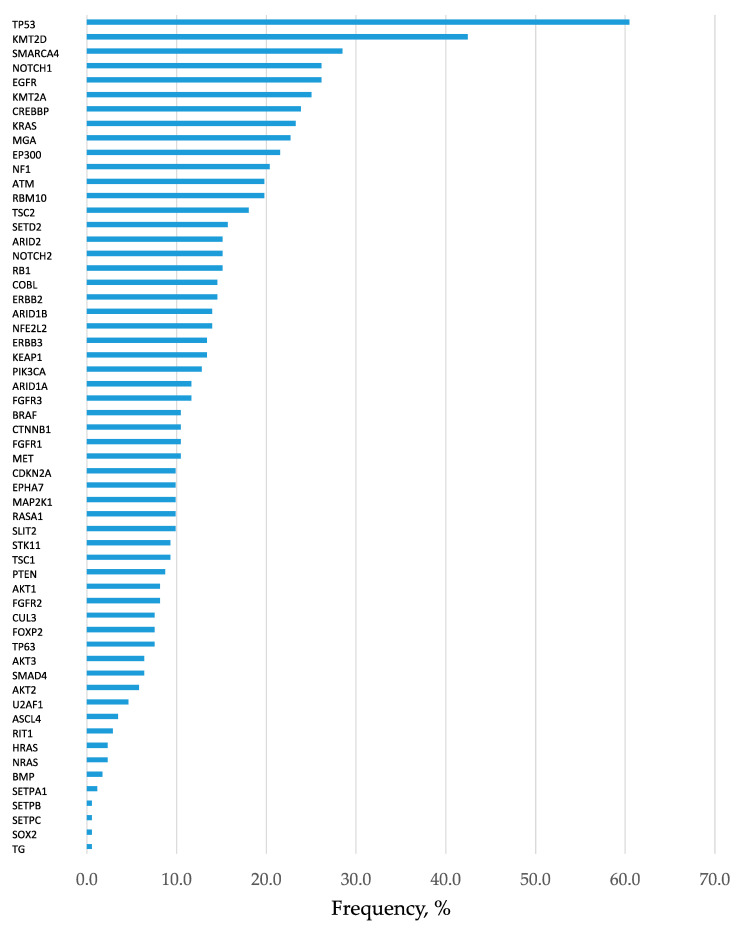
Frequency of detection of gene mutations. Analysis of all 172 patients. All mutations with allele fraction (AF) of >1% are presented in the order of frequency. Mutations in the tumor protein p53 (*TP53*) was the most common kind of mutation, present in approximately 60% of lung cancers.

**Figure 2 cancers-12-03472-f002:**
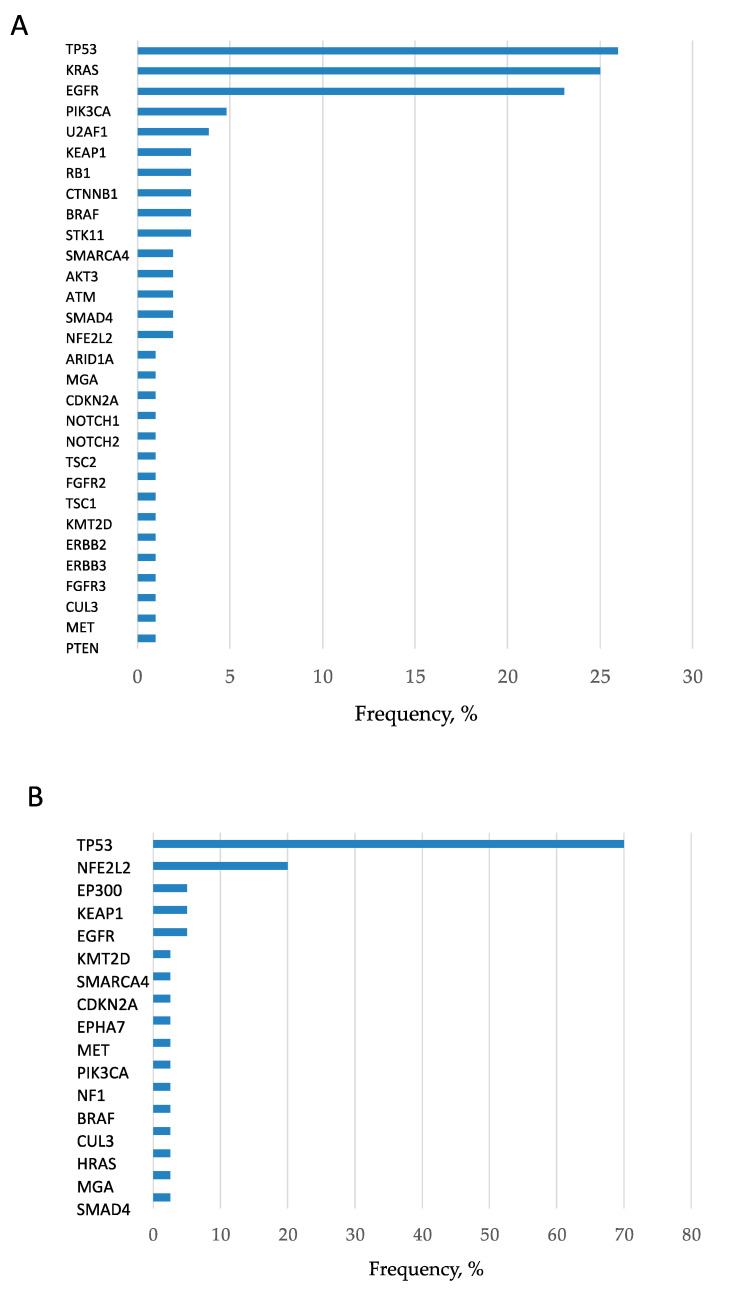
Frequency of detection of pathogenic mutations in adenocarcinomas and squamous cell carcinomas. (**A**) Pathogenic mutations with AF of >20 in adenocarcinomas are presented. The frequencies of *TP53*, *KRAS*, and epidermal growth factor receptor (*EGFR*) mutations were remarkably high, and hence they can be considered the three major mutations. (**B**) Pathogenic mutations with AF of >20 in squamous cell carcinomas are presented. The frequency of *TP53* mutation was remarkably high.

**Figure 3 cancers-12-03472-f003:**
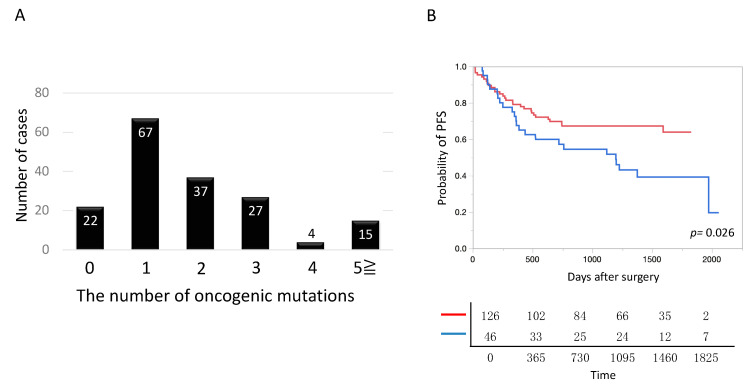
Number of pathogenic mutations in lung cancer. (**A**) The number of mutations harbored by each tumor is shown. Tumors harboring one mutation were most common. On average, tumors harbored 2.6 mutations. (**B**) The postoperative prognosis was significantly poorer in lung cancer, harboring three or more oncogenic mutations, than harboring 2 or less oncogenic mutations. Blue and red lines denote lung cancer harboring three or more oncogenic mutations (*n* = 46) and that harboring 2 or less oncogenic mutations (*n* = 126), respectively.

**Figure 4 cancers-12-03472-f004:**
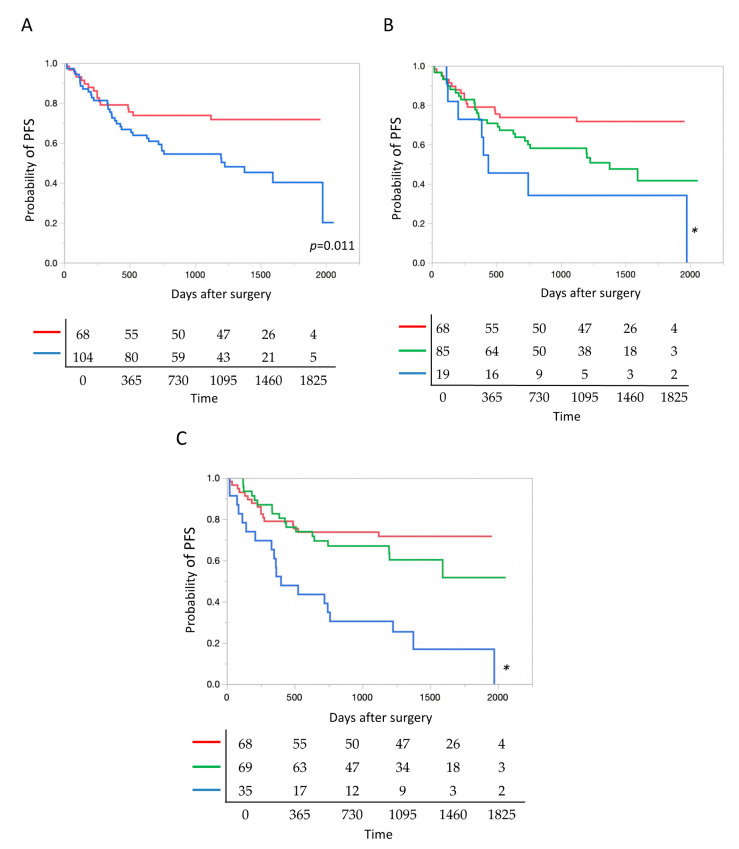
*TP53* mutation as a prognostic factor after surgery. (**A**) Postoperative prognosis was significantly poorer in lung cancer with tumor protein p53 (*TP53*) mutations than in lung cancer without *TP53* mutations. Blue and red lines denote cancers with, and without *TP53* mutations (*n* = 104 and 68), respectively. (**B**) The prognosis deteriorated significantly as the number of *TP53* mutations increased, i.e., in the ascending order of tumors with 0, 1, and 2 or more mutations. Red, green and blue lines denote cancers with 0, 1, and 2 or more mutations (*n* = 68, 85 and 19), respectively. * *p* = 0.014. (**C**) According to allele fractions (AFs) of *TP53* mutations, the prognosis of lung cancer was significantly poorer in the AF > 60 group than in the other two groups with AF = 0 and 0 < AF < 60. Blue, green and red lines denote cancers with the AF > 60, 0 < AF < 60, and AF = 0 (*n* = 35, 69 and 68), respectively. * *p* < 0.0001.

**Figure 5 cancers-12-03472-f005:**
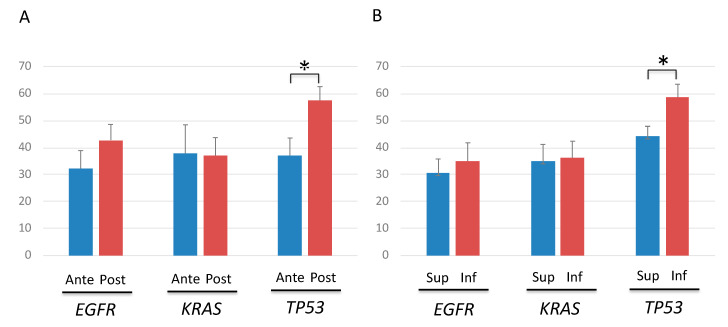
Allele fraction (AF) of *TP53, EGFR* and *KRAS* mutations according to the cancer location. (**A**) Comparison of AFs between tumors located at the ventral and dorsal sides showed significantly higher AFs only in tumor protein p53 (*TP53*)-mutated tumors located at the dorsal side. Blue and red lines denote anteriorly-, and posteriorly-located cancers, respectively. * *p* < 0.05. (**B**) Comparison of AFs between tumors located at the cranial and caudal sides showed significantly higher AFs only in *TP53*-mutated tumors located at the caudal side. Blue and red lines denote superiorly- and inferiorly-located cancers, respectively. * *p* < 0.05.

**Figure 6 cancers-12-03472-f006:**
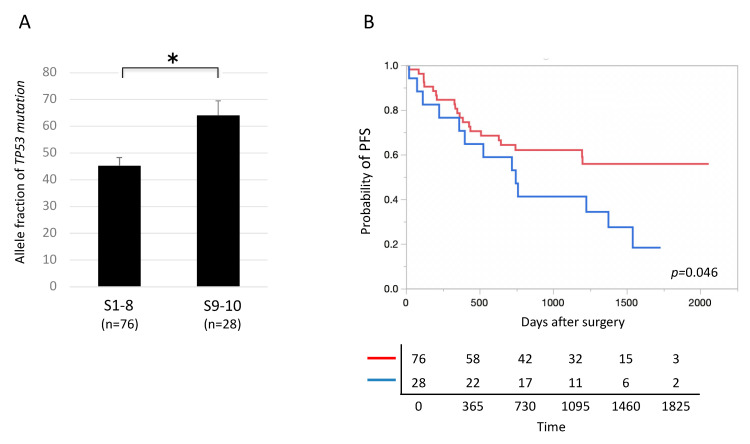
Locational variation in prognosis in lung cancers with *TP53* mutations. (**A**) Allele fractions of tumor protein p53 (*TP53*) mutations were significantly higher in *TP53*-mutated lung cancers located in segments 9 and 10 than in those located in segments 1 to 8. * *p* < 0.05. (**B**) The prognosis was significantly poorer in *TP53*-mutated lung cancers located in segments 9 and 10 than in those located in segments 1 to 8. Blue and red lines denote tumors located in segments 9–10 and 1–8 (*n* = 28 and 76), respectively.

**Table 1 cancers-12-03472-t001:** Patient characteristics.

Parameters	Variables	Total Number	Percentages
Total number		172	
Age (years), median (range)		71 (44–90)	
Sex			
	Male	116	67.4%
	Female	56	32.6%
Histology			
	Adeno	103	59.9%
	Squamous	40	23.3%
	Adenosquamous	4	2.3%
	Small	10	5.8%
	other	15	8.7%
Pathological stage			
	Ⅰ	79	45.9%
	Ⅱ	21	12.2%
	Ⅲ	33	19.2%
	IV	39	22.7%
Smoking status			
	Smoker	137	79.7%
	Non-smoker	35	20.3%

Adeno, adenocarcinoma; Squamous, squamous cell carcinoma; Adenosquamous, adenosquamous carcinoma; Small, small cell carcinoma.

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
