# Peer review of "Association of Mutation Profiles with Postoperative Survival in Patients with Non–Small Cell Lung Cancer"

_cancers, 2020, doi:10.3390/cancers12113472_

Round 1

Reviewer 1 Report

In this manuscript, the authors identify potential mutational patterns predicting prognostics in NSCLC. And they identified pathogenic TP53 mutations that could be correlated to poor prognosis of NSCLCs in particular anatomic regions. Their findings are quite interesting to me. But I still have some questions as following

  1. The authors need to list all the pathogenic variants in a supplementary table. The authors may refer to Mutation Annotation Format (MAF, https://docs.gdc.cancer.gov/Data/File_Formats/MAF_Format/).
  2. As mentioned in Reference 40, FATHMM-XF was developed for single-nucleotide variations (SNVs). However, the inframe deletions in exon 19 of EGFR are also important driver mutations. The author may need to check this kind of mutations included in their pathogenic variations or not.
  3. Most Kaplan-Meier plots demonstrated the trends that the mutational patterns could reflect prognostic status. However, it seems not an obvious difference between groups, especially within the first year. The differences between groups mostly became significant after one year. And seems the patient numbers (number at risk) became quite low. We may need to examine the results more carefully. The author may need to provide a table of "number at risk". They can group by year. And the authors also need to provide an actual p-value for each KM plot, not just less than 0.05. 
  4. It would be great to examine this algorithm with an independent dataset, such as TCGA-LUAD and TCGA-LUSC. The authors could use the pathogenic mutations, which are identified in this manuscript, to annotate TCGA-LUAD/LUSC samples. Then, process survival analysis.  
  5. Figure 1 and figure 2 loss of X-axial labeling. And please write down the actual p-value of each figure.

Author Response

  1. The authors need to list all the pathogenic variants in a supplementary table. The authors may refer to Mutation Annotation Format (MAF, https://docs.gdc.cancer.gov/Data/File_Formats/MAF_Format/).

Response: All pathogenic variants identified in our assay have been listed in Supplementary Table S1, per the reviewer’s suggestion.

  1. As mentioned in Reference 40, FATHMM-XF was developed for single-nucleotide variations (SNVs). However, the inframe deletions in exon 19 of EGFR are also important driver mutations. The author may need to check this kind of mutations included in their pathogenic variations or not.

Response: In addition to COSMIC database (FATHMM-XF), OncoKB database was utilized to estimate oncogenic function of indel mutations including inframe deletions in exon 19 of EGFR gene. OncoKB is a precision oncology knowledge base that annotates the oncogenic effects and clinical actionability of somatic alterations in cancer. 

We added this description to the Methods section.

  1. Most Kaplan-Meier plots demonstrated the trends that the mutational patterns could reflect prognostic status. However, it seems not an obvious difference between groups, especially within the first year. The differences between groups mostly became significant after one year. And seems the patient numbers (number at risk) became quite low. We may need to examine the results more carefully. The author may need to provide a table of "number at risk". They can group by year. And the authors also need to provide an actual p-value for each KM plot, not just less than 0.05. 

Response: The median (range) follow-up period for all censored cases was 973 (21–2056) days. Per the reviewer’s suggestion, the survival was re-analyzed using Jmp software, which confirmed the consistency in p-values, thereby indicating significant differences between the groups. Moreover, the number at risk was added to each survival curve, and the actual P values have been described in the figure legend and graph accordingly.

  1. It would be great to examine this algorithm with an independent dataset, such as TCGA-LUAD and TCGA-LUSC. The authors could use the pathogenic mutations, which are identified in this manuscript, to annotate TCGA-LUAD/LUSC samples. Then, process survival analysis.  

Response: It seems very interesting, and we hope to add this aspect to our future research study after considering the reviewer’s suggestion. However, we feel that the examination of TCGA-LUAD and TCGA-LUSC using the same methodology that has been currently used is beyond the scope of the current study.

We thank the reviewer for this valuable comment.

  1. Figure 1 and figure 2 loss of X-axial labeling. And please write down the actual p-value of each figure.

Response: The X-axis has been labelled in both Figure 1 and 2. Figure 1 and 2 represent the frequency of gene mutations and do not contain p-values.

I believe your comments have improved our manuscript so much. Thank you very much for your thoughtful comments.

Reviewer 2 Report

The manuscript “Association of mutation profiles with postoperative survival in patients with non–small cell lung cancer” by Goto et al describes the results of targeted sequencing analysis (53 genes) in 172 lung cancer patients. A good analysis of TP53 mutations in lung cancer is presented; particularly, regarding the different spatial distribution of the mutations. Pertinent figures and tables are included. However, the manuscript presents some critical flaws.

Major:

  1. The manuscript is confusing. The authors describe oncogenic and non-oncogenic mutations. It is not clear how those mutations have been defined, and how it has been determined their “pathogenic/oncogenic” (proliferative) role.
  2. The Methods section is very generic, and it should describe more detailed (e.g. describe the AF; describe the segment and all the bioinformatic analyses)
  3. The discussion should reflect the fact that only 53 genes have been analyzed (this is a limitation). Target sequencing does not “exhaustively” detect gene mutations. Many parts of the discussion should be clarified (e.g. mutation pathway/functional pathways of driver mutations). The authors should discuss some of the other manuscripts that have analyzed TP53 and prognosis in lung cancer.
  4. Have some of the mutations been validated with Sanger sequencing? If not, please indicate the parameters used to call a mutation.

Minor:

  1. A cartoon indicating the lung side segments would help the reader to understand the analysis
  2. Clarify if all the identified mutations are non-synonymous and whether they mapped in the exons (sometimes, the primers cover intronic regions)

Author Response

  1. The manuscript is confusing. The authors describe oncogenic and non-oncogenic mutations. It is not clear how those mutations have been defined, and how it has been determined their “pathogenic/oncogenic” (proliferative) role.

Response: Software that can automatically differentiate between oncogenic and non-oncogenic mutations is not available yet, and therefore, we analyzed each of the 2372 mutations against the OncoKB and COSMIC databases to annotate the mutations with possible oncogenic functions. Functional Analysis through Hidden Markov Models prediction in the COSMIC database was used to estimate oncogenic function of single nucleotide variations. In order to estimate oncogenic function of indel mutations, OncoKB, a comprehensive and curated precision oncology knowledge base, was utilized. Thus, all detected mutations were annotated by either COSMIC or OncoKB database.

              We added these descriptions in the Methods section. In addition, Supplementary Table 1 was added to the manuscript, which listed all the detected pathogenic mutations in our study.

  1. The Methods section is very generic, and it should describe more detailed (e.g. describe the AF; describe the segment and all the bioinformatic analyses)

Response: The allele fractions (AFs) represent the number of mutant reads divided by the total number of reads (coverage) at a specific genomic position. Explanatory schemes of the anatomical lung segments have been provided in Supplementary Figure 2. All the bioinformatic analyses were also described in more detail in the revised manuscript.

  1. The discussion should reflect the fact that only 53 genes have been analyzed (this is a limitation). Target sequencing does not “exhaustively” detect gene mutations. Many parts of the discussion should be clarified (e.g. mutation pathway/functional pathways of driver mutations). The authors should discuss some of the other manuscripts that have analyzed TP53 and prognosis in lung cancer.

Response: As the reviewer suggested, unlike whole exome sequencing, panel sequencing does not provide information regarding all exons, and therefore, it is likely possible that each tumor may actually possess much more pathogenic mutations. However, for the establishment of the cancer panel, we searched the literature and selected the 53 lung cancer-associated genes based on the following criteria: (a) genes often involved in lung cancer reported in The Cancer Genome Atlas and other projects or (b) genes frequently mutated in lung cancer in the Catalogue of Somatic Mutations in Cancer (COSMIC) database. Since the lung cancer panel that we developed covers the main hot spots of gene mutations in lung cancer, we assume that almost all SMGs (significantly mutated genes) associated with lung cancer can be detected. We added these descriptions in the Discussion section.Furthermore, we added some discussions in reference to previous studies that analyzed TP53 and prognosis in lung cancer.

  1. Have some of the mutations been validated with Sanger sequencing? If not, please indicate the parameters used to call a mutation.

Response:

In this study, we did not validate any detected mutations using Sanger sequencing. However, in some of our previous studies (doi: 10.1186/s12885-017-3059-1, doi: 10.3390/cancers11040552.), the mutations detected via panel sequencing were further validated using Sanger sequencing. Therefore, in this study, the data obtained using panel sequencing is considered to be scientifically reliable.

In this study, buffy coat DNA was used as a control to detect confident variants in tumors (Tumor–Normal pairs). We used the following filtering parameters for variant calling on Ion Reporter Local Server: (i) the minimum number of variant allele reads was ≥10, (ii) the coverage depth was ≥50, (iii) variant allele fraction (AF) ≥0.01, (iv) UCSC Common SNPs = Not In, (v) Confident Somatic Variants = In. 

We added these descriptions in the Methods section. 

Minor:

  1. A cartoon indicating the lung side segments would help the reader to understand the analysis

Response: Per the reviewer’s suggestion, explanatory schemes of the anatomical lung segments have been provided in Supplementary Figure 2.

  1. Clarify if all the identified mutations are non-synonymous and whether they mapped in the exons (sometimes, the primers cover intronic regions)

Response: In the cancer panel, the primers were designed to cover the hot-spot mutations present in the lung cancer-associated genes. Therefore, this targeted sequencing method exclusively detects non-synonymous mutations present in the exons.

We added these descriptions in the Methods section. 

Thank you very much for your thoughtful comments.

Reviewer 3 Report

Goto et al. present a well conducted study that, taking advantage of NGS technology, analyzes the mutations in 53 lung cancer-associated genes of 172 surgically resected lung cancers and associates the genetic profiles with clinical outcomes.

The authors found that the number of mutations, the presence of a TP53 somatic mutation, and a high TP53 allele fraction were associated with a worse prognosis. Although similar studies with similar conclusions have been presented previously in the literature using other methodologies, the application of targeted NGS in this study provides an added dimension. In addition, this study shows that the TP53 mutation profile varies with the site of lung tumor. Overall, there is no major innovation in the results herein presented, nevertheless, the authors did a commendable work. However, a few points should be better clarified, and some considerations should be made before being accepted for publication, as follows:

  1. Molecular tests routinely performed for lung cancers such as ROS1 and ALK and possibly RET, are not reported and should be included in this study.
  2. Since the genetic profile can be affected by therapy, it should be reported if the patients received preoperative therapy, and which.
  3. On the paper, any found mutation is referred to as “driver”. The concept of driver mutation refers to mutations that confer a fitness advantage to somatic cells as opposed to passenger mutations. The author should clarify on which basis all somatic mutation identified have been considered as driver.
  4. The discussion does not relate the study findings to those of previous studies. Moreover, the authors do not mention and discuss many pertinent studies on similar issues in the last 5 years, such as:

Genetic alterations of driver genes as independent prognostic factors for disease-free survival in patients with resected non-small cell lung cancer. Lung Cancer 128 (2019) 152-157

Prognostic and Predictive Effect of TP53 Mutations in Patients with Non-Small Cell Lung Cancer from Adjuvant Cisplatin-Based Therapy Randomized Trials: A LACE-Bio Pooled Analysis J Thorac Oncol. 2016 Jun;11(6):850-61

Multiregion sequencing reveals the intratumor heterogeneity of driver mutations in TP53-driven non-small cell lung cancer. Int J Cancer. 2017 Jan 1;140(1):103-108.

The influence of TP53 mutations on the prognosis of patients with early stage non-small cell lung cancer may depend on the intratumor heterogeneity of the mutations.  Mol Carcinog. 2015 Feb;54(2):93-101.

Tumor Mutation Burden as a Biomarker in Resected Non-Small-Cell Lung Cancer. J Clin Oncol. 2018 Oct 20;36(30):2995-3006

Tracking the Evolution of Non-Small-Cell Lung Cancer. N Engl J Med. 2017 Jun 1;376(22):2109-2121.

The Impact of Smoking and TP53 Mutations in Lung Adenocarcinoma Patients with Targetable Mutations-The Lung Cancer Mutation Consortium (LCMC2). Clin Cancer Res. 2018 Mar 1;24(5):1038-1047.

A novel tumor mutational burden estimation model as a predictive and prognostic biomarker in NSCLC patients. BMC Med. 2020 Aug 26;18(1):232.

Author Response

  1. Molecular tests routinely performed for lung cancers such as ROS1 and ALK and possibly RET, are not reported and should be included in this study.

Response: Fusion gene screening through transcriptome sequencing requires assays that are different from DNA mutation analysis. When we performed a different assay (Oncomine® Platform) to search for ROS1, ALK, and RET fusion genes in the 172 patients who enrolled in this study, these genes were not detected in any of the patients. These descriptions were added to the Results section.

  1. Since the genetic profile can be affected by therapy, it should be reported if the patients received preoperative therapy, and which.

Response: One patient received four cycles of preoperative cisplatin+vinorelbine chemotherapy. This description has been added to the Results section. Since there was only one patient, we could not examine or discuss the mutational changes (modifications) induced due to chemotherapy.

  1. On the paper, any found mutation is referred to as “driver”. The concept of driver mutation refers to mutations that confer a fitness advantage to somatic cells as opposed to passenger mutations. The author should clarify on which basis all somatic mutation identified have been considered as driver.

Response: Only pathological mutations at an AF of >20% were regarded as driver mutation in our study. We added this description in the section 2.2. In the Discussion section, the terms “driver mutation” were corrected to “pathogenic mutation”.

  1. The discussion does not relate the study findings to those of previous studies. Moreover, the authors do not mention and discuss many pertinent studies on similar issues in the last 5 years, such as:

Genetic alterations of driver genes as independent prognostic factors for disease-free survival in patients with resected non-small cell lung cancer. Lung Cancer 128 (2019) 152-157

Prognostic and Predictive Effect of TP53 Mutations in Patients with Non-Small Cell Lung Cancer from Adjuvant Cisplatin-Based Therapy Randomized Trials: A LACE-Bio Pooled Analysis J Thorac Oncol. 2016 Jun;11(6):850-61

Multiregion sequencing reveals the intratumor heterogeneity of driver mutations in TP53-driven non-small cell lung cancer. Int J Cancer. 2017 Jan 1;140(1):103-108.

The influence of TP53 mutations on the prognosis of patients with early stage non-small cell lung cancer may depend on the intratumor heterogeneity of the mutations.  Mol Carcinog. 2015 Feb;54(2):93-101.

Tumor Mutation Burden as a Biomarker in Resected Non-Small-Cell Lung Cancer. J Clin Oncol. 2018 Oct 20;36(30):2995-3006

Tracking the Evolution of Non-Small-Cell Lung Cancer. N Engl J Med. 2017 Jun 1;376(22):2109-2121.

The Impact of Smoking and TP53 Mutations in Lung Adenocarcinoma Patients with Targetable Mutations-The Lung Cancer Mutation Consortium (LCMC2). Clin Cancer Res. 2018 Mar 1;24(5):1038-1047.

A novel tumor mutational burden estimation model as a predictive and prognostic biomarker in NSCLC patients. BMC Med. 2020 Aug 26;18(1):232.

Response: Upon referring to the papers suggested by the reviewer, some aspects have been elaborated in the Discussion section of the manuscript as follows:

“Regarding the prognostic effects of TP53 mutations on non-small cell lung cancer detected via direct sequencing, inconsistent results have been reported previously. Aisner et al. reported that TP53 mutations are associated with shorter survival, whereas Ma et al. and Devarakonda et al. reported that TP53 mutations exhibited no such prognostic effect. Meanwhile, some other studies have revealed that TP53 mutations frequently exhibit intratumor heterogeneity (ITH) in NSCLC. Based on this finding, Lee et al. reported that TP53 mutations with ITH exhibit worse survival when compared to wild-type TP53, whereas TP53 mutations without ITH did not exhibit such an extreme phenotype.”

Thank you very much for your thoughtful comments.

Round 2

Reviewer 2 Report

The authors have responded satisfactorily  to the comments.

Reviewer 3 Report

My concerns have been addressed satisfactorily.